# Antibiotic consumption and medication cost in diabetic patients: Insights from Iran health insurance organization (IHIO) claims data

Arash Bagherian Ghotbi[1‡], Benyamin Khoshparast[1‡], Hamidreza Hekmat[2], Zahra Shahali[3], Ali Golestani[1]*, Ozra Tabatabaei-Malazy[4]*

**1** Non-Communicable Diseases Research Center, Endocrinology and Metabolism Population Sciences Institute, Tehran University of Medical Sciences, Tehran, Iran, **2** School of Medicine, Baharloo Hospital, International Campus, Tehran University of Medical Sciences, Tehran, Iran, **3** National Center for Health Insurance Research, Tehran, Iran, **4** Endocrinology and Metabolism Research Center, Endocrinology and Metabolism Clinical Sciences Institute, Tehran University of Medical Sciences, Tehran, Iran

‡ These authors contributed equally to this work as the first authors on this work.
* aligolestani7597@gmail.com (AG); tabatabaeiml@sina.tums.ac.ir (OTM)

## Abstract

### Background

The rising prevalence of diabetes is increasing the healthcare costs especially when associated with infection. We aimed to assess the antibiotic consumption and medication costs in diabetes.

### Methods

We performed a retrospective claims-based study using Iranian Health Insurance Organization (IHIO) dataset from 24 provinces during 2014–2017. Systemic antibacterials were quantified in defined daily doses and diabetic patients were stratified into "No antibiotic" (NAb) and quartiles of cumulative antibiotic exposure (Q1–Q4). A dominant antidiabetic regimen was assigned when ≥80% of a patient's diabetes prescriptions came from one drug class or combination. Inflation-adjusted annual medication costs were modelled with log-link Gamma generalized linear models.

### Results

The study comprised 1,704,182 individuals (62.0% women). Biguanides alone were most common dominant diabetes regimen (40%), whereas penicillin accounted for 35.8% of all antibiotic dispensing. Mean annual medication costs were 93 USD for women and 138 USD for men; however, after adjustment men incurred slightly lower costs than women. Compared with the NAb group, costs rose progressively with antibiotic exposure, reaching an adjusted mean ratio (MR) 3.17 (95%CI 3.09–3.25) in Q4. Relative to biguanide monotherapy, costs were markedly higher for regimens biguanides + insulins (MR 5.75, 5.54–5.97) or insulins alone (MR 5.53, 5.38–5.68).

**Data availability statement:** These datasets presented in this article are not readily available because they are obtained from the IHIO database, and due to privacy concerns for the patients, authors are not permitted to share the data publicly or privately. However, any researcher with written permission can request to obtain the anonymized data. Requests for access to a de-identified dataset may be sent to IHIO Data Access Committee (https://nchir.ihio.gov.ir/).

**Funding:** This research has been supported by Tehran University of Medical Sciences & health services grant numbered 1402-3-221-68891. The funder had no role in any part of study, design, data collection, analysis, interpretation or writing. OTM received the award.

**Competing interests:** The authors have declared that no competing interests exist.

**Abbreviations:** ATC, Anatomic Therapeutic Chemical; DDD, Defined Daily Doses; DM, Diabetes Mellitus; EMRO, Eastern Mediterranean Region; GLM, Gamma Generalized Linear Model; IHIO, Iranian Health Insurance Organization.

## Conclusion

Quantifying the joint impact of antidiabetic regimens and antibiotic use on treatment costs highlights key factors driving healthcare expenditures. These findings can inform targeted antibiotic stewardship strategies and guide reimbursement policy to optimize resource allocation and reduce the financial burden on both patients and insurers.

## Introduction

Diabetes mellitus (DM) is a major lifestyle-disrupting disease with substantial economic burden and a rapidly increasing global prevalence [1,2]. In 2021, an estimated 537 million people were living with diabetes worldwide, and this number is projected to rise by 46% by 2045 [1]. In Iran, prevalence among adults nearly doubled between 2007 and 2021, reaching approximately 14% in 2021, amid trends in urbanization, sedentary lifestyles, unhealthy behaviors, and population aging; over the same period, the quality of diabetes care has shown a decline [3,4]. The financial impact is considerable: global diabetes-related healthcare costs for individuals aged 18–99 are predicted to approach $985 million by 2045, and in Iran, per-person expenditures were about $1,300 in 2021 with projections of $1,800 by 2045 [5].

Beyond chronic complications, individuals with DM are more susceptible to infections across multiple organ systems, including skin and soft tissues, respiratory tract, and urinary tract, involving both common and less typical pathogens [6,7]. Infection-related costs are substantial; diabetic foot complications alone drive high expenditures for infected ulcers and amputations, and infections increase healthcare utilization through hospital admissions and prolonged treatments, placing additional burdens on health systems and insurers [8–11]. These pressures underscore the importance of preventive care and aligning reimbursement policies with best practices, particularly for diabetes-related foot disease [9].

Despite rising diabetes prevalence in Iran, there are critical gaps in real-world evidence on antidiabetic and antibiotic prescribing patterns and their cost implications among people with diabetes. In particular, antibiotic use patterns in this vulnerable population remain poorly characterized. Using pharmacy claims from the Iran Health Insurance Organization (IHIO) across 24 provinces (2014–2017), this study aims to quantify anti-diabetic and antibiotic consumption in different antibiotic use groups and to assess the annual medication costs in adults with DM. By illuminating prescribing and cost patterns, our goal is to inform strategies that balance therapeutic efficacy, cost containment, and patient safety within Iran's health insurance context.

## Materials and methods

### Study population

This retrospective study utilized claims data from the IHIO, collected from pharmacies which had contracts with this insurance organization across 24 of Iran's 31 provinces between March 21, 2014, and March 20, 2017. The unavailability of the data on some

provinces including Ardabil, Alborz, East Azerbaijan, West Azerbaijan, Khuzestan, Qom, and Semnan was due to the incompleteness of provincial-level collection system in aforementioned provinces at the time. The IHIO is a public organization operating under the supervision of the Ministry of Health and Medical Education (MoHME) and covers nearly 50% of the Iranian population [12]. The main dataset covered approximately 19 million individuals who took at least one prescription during study period. Researchers were granted access to the fully anonymized data in December 2023, at which point analysis began. The data used in this study were anonymized, and no identifying information (e.g., name or national code) was accessible to the authors. Patients aged 18–95 years old with at least one prescription for diabetes treatment and without any missing values for other variables in the dataset were included in this study. Based on a previous study estimating the prevalence of diabetes at national level in Iran in 2016, approximately 5.2 million people had diabetes [13]. Considering our study consisted of about 1.7 million patients with diabetes, the study population represented roughly 32% of all individuals with diabetes in the country.

This study had two main outcomes. First, it described the prescription patterns of antidiabetic and antibiotic medications across different groups of antibiotic use. Second, it compared the costs of antidiabetic medications, antibiotics, and other drugs across available sociodemographic and clinical variables.

## Data source variables

The IHIO pharmacy claims database provides details on dispensed medications for each individual within the scheme. Medications are coded using the World Health Organization (WHO) Anatomical Therapeutic Chemical (ATC) classification system [14] and prescriber information, Defined Daily Doses (DDD), strength, quantity, method and unit of administration of each drug dispensed, ingredient costs are available. The ATC classification system, recommended by the WHO, categorizes drugs based on their therapeutic use, pharmacological properties, chemical characteristics, and the target organ or system [14]. The DDD is a standardized metric for drug consumption, representing the assumed average daily maintenance dose for a drug's primary indication in adults. This system facilitates the comparison of drug use across different medications and healthcare settings [15].

Information on sex, date of birth, date of claim, province of claim, claim costs (including out-of-pocket and total costs), and health insurance fund membership is recorded for each claimant, while diagnostic data and outcomes are not available. The date of claim was used to determine the month and season of each prescription. The IHIO encompasses several major funds including Rural, Civil Servants, Iranian, Universal, Foreign, and "Other funds". The Civil Servants fund provides coverage for all civil servants, including those who are employed, retired, or receiving pensions. The Rural fund serves villagers, nomadic populations, and residents of towns with fewer than 20,000 inhabitants. The Iranian and Universal Health Insurance funds are accessible to all Iranian citizens either through full premium contributions or based on household income. The Foreign fund, designed for non-Iranian nationals; and the "Other funds", which insures veterans, war-injured individuals, students, people with disabilities, welfare beneficiaries, prisoners, and their families [16].

## Data pre-processing

Data curation and preparation in this study were mainly straightforward. However, one notable challenge involved missing values for the medication dosage variable, requiring manual extraction of dosage information from the full medication names. Patient age was calculated by subtracting the date of birth from the date of prescription dispensation, and the mode of the resulting values was used as the patient's age during the study. Individuals younger than 18 or older than 95 were excluded. Patients were also categorized by age group, and their province was determined by identifying the most frequently listed province in their prescriptions.

## Identifying patients with diabetes

Patients were identified as diabetic if they had at least one prescription for either "insulins and analogues" (A10A) or "blood glucose-lowering drugs, excluding insulins" (A10B). Using prescription drug data to identify chronic diseases is a

validated approach in population-level studies when administrative data lack standardized diagnostic codes [17–19]. However, it is prone to overestimation because it has high specificity but limited sensitivity [20]. Only individuals with diabetes were included in the analysis. Antibiotics considered in this study were those categorized as "antibacterials for systemic use" (J01) under the ATC system. The analysis also included reports on subclasses within the A10B and J01 codes.

## Determining antibiotic use groups

To determine the duration of antibiotic use, the DDD was used in conjunction with dosage and quantity information. A 30-day threshold was used to identify outliers, and single antibiotic prescriptions exceeding this duration were excluded as invalid. Patients were categorized into groups based on the sum of their antibiotic prescription durations. The "No antibiotic" (NAb) group included those without any antibiotic prescriptions during the study period, while the remaining patients were divided into four quartiles (denoted as Q1, Q2, Q3, and Q4) based on their cumulative duration of antibiotic use, with cut-off points rounded to the nearest whole day.

## Determining dominant diabetes treatment regimen

To determine each patient's dominant diabetes treatment regimen, we used prescription-level data for the eight classes of glucose-lowering medications based on the ATC classification. For each patient, the total number of prescriptions in these classes was summed, and the relative contribution of each drug class was calculated as a proportion of the total. We then ranked each patient's drug classes in descending order based on prescription volume and calculated the cumulative proportion of use across classes. A regimen was classified as "dominant" if one or more drug classes together accounted for at least 80% of that individual's total glucose-lowering prescriptions. The combination of drug classes that crossed this 80% threshold was recorded as the patient's dominant regimen. For example, if biguanides alone comprised ≥80% of prescriptions, the regimen was classified as "biguanides alone"; if both biguanides and insulins together exceeded the threshold, the regimen was noted as a combination. Insulin use was also flagged as a binary variable, identifying patients who had received any prescriptions for insulins or analogues (A10A), regardless of whether insulin was part of the dominant combination. The resulting classifications were stored for further statistical analysis.

## Cost analysis

To ensure comparability and interpretability of costs, they were adjusted for health inflation to match the values of the first study year. Then the exchange rate of USD to Rials for the first year of the study (1 USD = 25,942 Rials) was used to report the costs. For patients whose first diabetes drug prescription occurred in the second or third year of the study, the diabetes-related portion of their costs was calculated by dividing their diabetes drug costs by the number of years since they began treatment, ensuring a more representative cost estimate. All costs reported and modeled in this study are annual. The median and mean annual medication cost per individual, as well as the proportions attributed to diabetes medications, antibiotics, and other medications, were calculated, with interquartile ranges and confidence intervals provided. Additionally, the proportion of out-of-pocket costs was reported.

## Statistical analysis

Descriptive statistics, including means, medians, interquartile ranges (IQR), frequencies, percentages, proportions, and 95% confidence intervals (95%CI), were calculated for demographic variables, drug classes, and dominant diabetes treatment regimens across the different groups. Gamma Generalized Linear Models (GLM) were employed to model the total medication costs of diabetic patients. These are the preferred models for right-skewed economic variables, such as healthcare costs and hospital length of stay, due to their superior performance in terms of parameter bias, standard errors, and predictive accuracy compared to traditional models [21]. The total medication costs were modeled using Gamma GLMs, both for each variable individually (crude model) and with all variables adjusted simultaneously. Mean ratios, and

p-values for the Gamma GLM coefficients were reported, where the mean ratio represents the expected value mean ratio of the outcome variable under different conditions or groups. All analyses were conducted using Python version 3.11.4, utilizing the libraries stats models, pandas, and numpy.

### Ethical statement

This study adhered to the Declaration of Helsinki and received ethical approval from the Research Ethics Committee of Endocrine & Metabolism Research Institute, Tehran University of Medical Sciences (Approval ID: IR.TUMS.EMRI. REC.1402.094). IHIO provided fully anonymized data before investigator access.

## Results

### Description of included individuals and prescriptions

A total of 1,704,182 individuals with diabetes mellitus were included in the study, with 99,859,824 prescriptions recorded (Table 1 and Table 2). The highest population prevalence, across groups NAb to Q4, was in the 40–64 age group, while the lowest was in the 65–95 age group, with a decreasing trend from NAb to Q4 (24.17% to 18.85%) in the older group. Additionally, 66.94% of the study population were women, and 33.06% were men. As we moved from NAb to Q4, the proportion of women increased from 62.01% to 68.82%. Also, Civil Servants insurance saw an increasing trend (26.12% to 50.93%), while the Rural insurance group showed a decrease (39.35% to 6.48%) across the antibiotic groups (Table 1). Percentages within each demographic and insurance stratifications sum to 100%, and are presented with 95% confidence intervals. The percentage of included participants based on provinces are presented in S1 Table. Each antibiotic group included approximately 20% of the study population. There was an increasing trend in total prescriptions from the "No antibiotic" (NAb) group (6,426,526) to the Q4 group (40,929,474), showing a 6.36-fold increase (Table 2).

**Table 1. The percentage of included individuals based on demographic and insurance stratifications in the study.**

| variable | | No antibiotic | Q1 | Q2 | Q3 | Q4 | All |
|---|---|---|---|---|---|---|---|
| **Age group (Percentage (95% confidence interval))** | **18-39** | 25.34 (25.19-25.49) | 26.32 (26.17-26.46) | 26.24 (26.1-26.39) | 26.81 (26.65-26.96) | 25.34 (25.19-25.48) | 26.02 (25.96-26.09) |
| | **40-64** | 50.49 (50.32-50.67) | 49.84 (49.68-50.0) | 50.96 (50.8-51.12) | 52.03 (51.86-52.2) | 55.82 (55.65-55.98) | 51.82 (51.75-51.9) |
| | **65-95** | 24.17 (24.01-24.32) | 23.84 (23.7-23.98) | 22.8 (22.66-22.94) | 21.17 (21.03-21.31) | 18.85 (18.72-18.98) | 22.16 (22.1-22.22) |
| **Sex (Percentage (95% confidence interval))** | **Male** | 37.99 (37.82-38.16) | 34.23 (34.07-34.38) | 32.06 (31.9-32.21) | 30.18 (30.02-30.34) | 31.18 (31.02-31.33) | 33.06 (32.99-33.13) |
| | **Female** | 62.01 (61.84-62.18) | 65.77 (65.62-65.93) | 67.94 (67.79-68.1) | 69.82 (69.66-69.98) | 68.82 (68.67-68.98) | 66.94 (66.87-67.01) |
| **Fund (Percentage (95% confidence interval))** | **Civil servants** | 26.12 (25.97-26.28) | 37.11 (36.95-37.27) | 44.42 (44.26-44.58) | 49.5 (49.33-49.67) | 50.93 (50.76-51.1) | 41.83 (41.76-41.9) |
| | **Rural** | 39.35 (39.18-39.52) | 23.6 (23.46-23.74) | 16.37 (16.24-16.49) | 11.4 (11.29-11.51) | 6.48 (6.4-6.56) | 19.14 (19.08-19.2) |
| | **Iranian** | 11.45 (11.34-11.57) | 14.16 (14.05-14.28) | 14.73 (14.61-14.84) | 14.86 (14.74-14.98) | 12.31 (12.2-12.42) | 13.55 (13.5-13.6) |
| | **Universal** | 16.95 (16.82-17.08) | 17.07 (16.95-17.2) | 14.51 (14.4-14.63) | 11.38 (11.27-11.49) | 6.17 (6.08-6.25) | 13.21 (13.16-13.26) |
| | **Others** | 5.75 (5.67-5.83) | 7.85 (7.77-7.94) | 9.89 (9.79-9.99) | 12.83 (12.71-12.94) | 24.11 (23.96-24.25) | 12.13 (12.08-12.18) |
| | **Foreign** | 0.37 (0.35-0.4) | 0.21 (0.19-0.22) | 0.09 (0.08-0.1) | 0.03 (0.02-0.04) | 0.0 (0.0-0.01) | 0.14 (0.13-0.14) |
| **Total (Number)** | | 311,685 | 359,121 | 358,920 | 332,538 | 341,918 | 1,704,182 |

**Table 2. Percentage of prescriptions including antibiotics and glucose-lowering drugs based on different quartiles of antibiotics use.**

|  | All | No Antibiotic | Q1 | Q2 | Q3 | Q4 |
|---|---|---|---|---|---|---|
| Antibiotic (Percentage (95% confidence interval)) | 7.47 (7.46-7.47) | 0.00 (0.00-0.00) | 3.95 (3.94-3.97) | 5.97 (5.96-5.98) | 7.73 (7.72-7.74) | 10.22 (10.21-10.23) |
| Glucose-lowering drugs (Percentage (95% confidence interval)) | 7.79 (7.79-7.80) | 14.54 (14.51-14.57) | 11.19 (11.17-11.21) | 9.34 (9.33-9.36) | 7.67 (7.66-7.68) | 5.08 (5.08-5.09) |
| Total (Number) | 99,859,824 | 6,426,526 | 12,704,037 | 17,543,904 | 22,255,883 | 40,929,474 |

## Anti-diabetic medications prescription patterns

Of all prescriptions, 7.79% contained anti-diabetic drugs, showing a decreasing trend in their proportion from NAb to Q4 (14.54% to 5.08%). However, the total number of anti-diabetic drugs increased from NAb to Q4 (1,416,989–3,037,953 by 2.14 times) (Table 2). The most dominant anti-diabetic regimen across all groups (NAb to Q4) was Biguanides alone, comprising approximately 40% of the regimens (Fig 1 and S2 Table). The next most common regimens were Biguanides combined with Sulfonylureas (about 20%) and Insulin and its analogues (about 10%). The Q4 group had the highest proportion of Insulins and Analogues prescriptions (25.89%) and the lowest proportion of Biguanides prescriptions (42.54%) (Table 3). Conversely, this finding was reversed in Q2 (23.14% for Insulins and 43.92% for Biguanides). Two anti-diabetic drug classes, Thiazolidinediones and GLP-1 analogues, showed a decreasing trend in usage from NAb to Q4. However, this continuous trend was not observed in other drug classes.

## Antibiotic medications prescription patterns

Of all prescriptions, 7.47% contained antibiotic drugs, with an increasing trend in their proportion from Q1 to Q4 (3.95% to 10.22%) as the total number of antibiotic drugs increased (564,776–5,204,806 by 9.21 times) from 2014 to 2017 (Table 2). All antibiotic drug classes showed an increasing trend in usage from Q1 to Q4 groups, except for Penicillins, Macrolides, Lincosamides, Streptogramins, and Quinolones which showed a decreasing trend, and Sulfonamides and Trimethoprim, which did not show a continuous trend (Table 3). In all groups (Q1 to Q4), the highest proportion of antibiotics was devoted to Penicillins (35.75%), and the lowest to Amphenicols, with nearly 0%.

Regarding seasonal changes in antibiotic use, it was observed that usage consistently decreased after winter and increased after summer, in the three most used antibiotic classes: Penicillins, other beta-lactams, and macrolides, Lincosamides, and Streptogramins (Fig 2 and S3 Table). In contrast to these groups, quinolones exhibited an increasing trend from spring to summer. From 2014 to 2017, the range of fluctuations in antibiotic usage decreased. The highest antibiotic usage during the study period was recorded in winter 2015.

## Cost of medications

The mean annual expenditure for the people in this study was 93 USD for women and 138 USD for men (Table 4). However, the median of annual total costs was 29.21 USD for women and 30.88 USD for men (Table 5). Although the mean annual total costs were higher for men (45 USD more), the percentage of out-of-pocket payment was higher for women (19.37% for women and 18.16% for men). In both sexes, the "other drugs" had the highest proportion (55.92% for women and 54.2% for men) and antibiotics had the lowest proportion of the mean annual total costs (4.31% for women and 3.94% for men). However, out-of-pocket proportion of the payments was more for antibiotics (29.7% in average) in comparison to diabetic drugs (13.9% in average) and "other drugs" (21.55% in average). As age increased, the mean annual total expenditure also rose from 60 to 125 USD. In the 65–95 years old age group, the median cost was 47.4 USD. This amount was 17.58 for 18–39 years old. Also, there was an increasing trend in the percentage of the mean out-of-pocket expenditures from 18.72% to 19.86% on average (from 11.23 to 24.82 USD

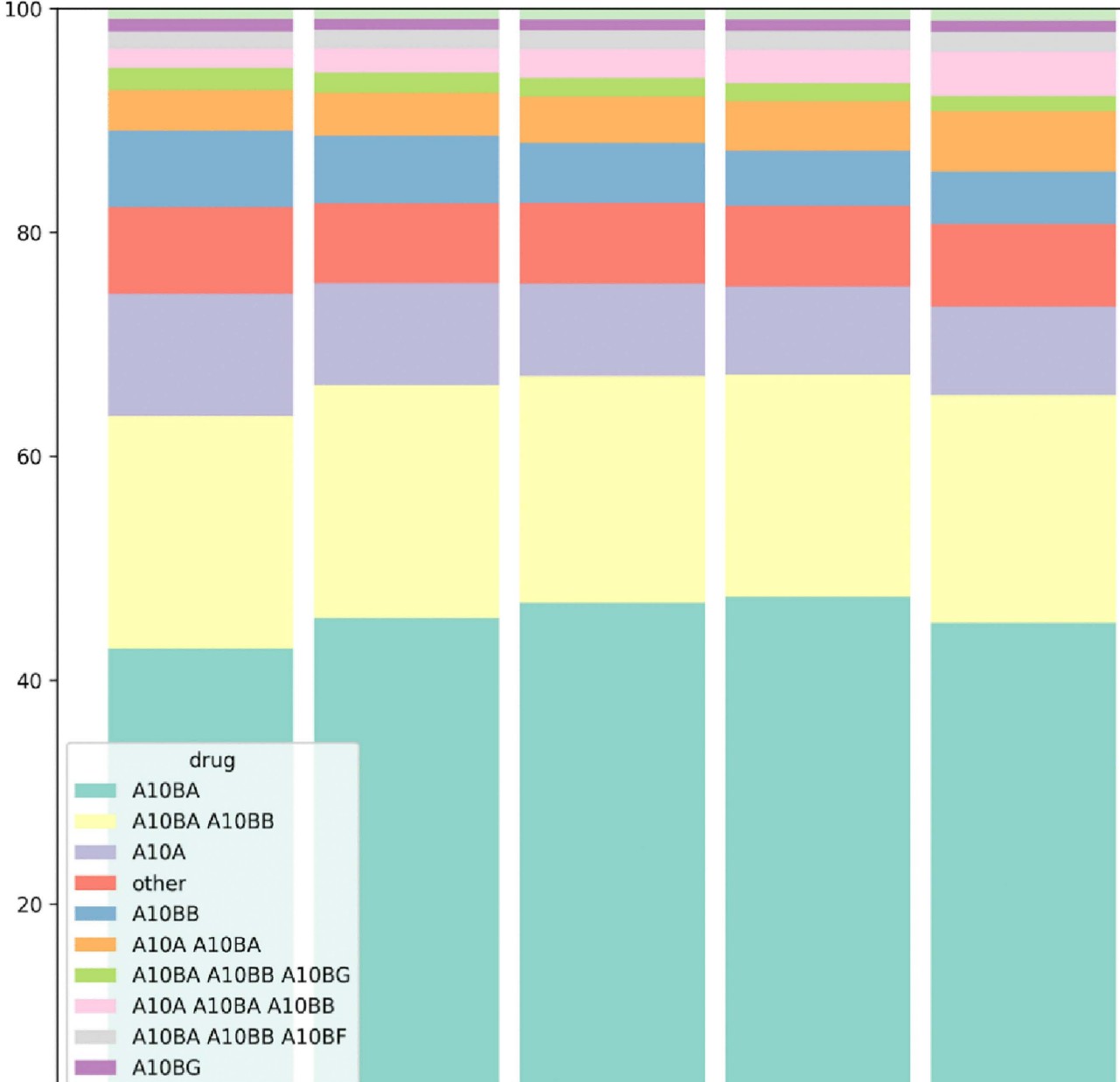

**Fig 1. Percentage of dominant diabetes treatment regimen within each antibiotic group.** A10A: Insulins and Analogues, A10BA: Biguanides, A10BB: Sulfonylureas, A10BD: Combinations of oral blood glucose lowering drugs, A10BF: Alpha glucosidase inhibitors, A10BG: Thiazolidinediones, A10BJ: Glucagon-like peptide-1 (GLP-1) analogues, A10BX: Other blood glucose lowering drugs, excluding insulins.

on average). Also, from NAb to Q4, the proportion of out-of-pocket payment had an increasing trend (from 17.6% to 19.92% of the total costs). In the provinces included in the study, Yazd had the highest (131 USD) and Lorestan had the lowest (69 USD) mean annual costs (S4 and S5 Tables). Regarding the insurances included in the study, patients with "Foreign" insurance had the lowest expenditure (37 USD) and patients with Civil Servants insurance had the highest expenditure (121 USD). The lowest out-of-pocket payment was for patients with Foreign Citizens insurance (4.99 USD) and the highest was for those who had "Other" insurance (25.77 USD).

**Table 3. Percentage of glucose-lowering and antibiotic drug class prescriptions within each antibiotic group.**

| Drugs, Class | | Antibiotic group | | | | |
|---|---|---|---|---|---|---|
| | | No Antibiotic | Q1 | Q2 | Q3 | Q4 |
| Glucose lowering drugs (Percentage (95% confidence interval)) | A10A | 24.67 (24.59-24.74) | 23.24 (23.18-23.30) | 23.14 (23.09-23.20) | 23.67 (23.62-23.72) | 25.89 (25.84-25.94) |
| | A10BA | 42.68 (42.59-42.76) | 43.71 (43.64-43.77) | 43.92 (43.86-43.99) | 43.81 (43.75-43.87) | 42.54 (42.49-42.60) |
| | A10BB | 22.40 (22.33-22.47) | 23.05 (23.00-23.11) | 22.89 (22.84-22.95) | 22.64 (22.59-22.69) | 22.51 (22.46-22.56) |
| | A10BD | 0.40 (0.39-0.41) | 0.29 (0.29-0.30) | 0.27 (0.26-0.28) | 0.30 (0.29-0.31) | 0.32 (0.31-0.33) |
| | A10BF | 3.31 (3.28-3.33) | 3.38 (3.36-3.41) | 3.59 (3.57-3.61) | 3.58 (3.56-3.61) | 3.41 (3.39-3.43) |
| | A10BG | 4.47 (4.44-4.50) | 4.31 (4.28-4.34) | 4.21 (4.19-4.24) | 4.13 (4.11-4.15) | 3.68 (3.66-3.71) |
| | A10BJ | 0.00 (0.00-0.00) | 0.00 (0.00-0.00) | 0.00 (0.00-0.00) | 0.00 (0.00-0.00) | 0.00 (0.00-0.00) |
| | A10BX | 2.08 (2.06-2.11) | 2.01 (1.99-2.03) | 1.97 (1.95-1.98) | 1.86 (1.84-1.88) | 1.64 (1.63-1.65) |
| | Total glucose lowering drug prescriptions (Number) | 1,416,989 | 2,133,477 | 2,447,351 | 2,534,071 | 3,037,953 |
| Antibiotics (Percentage (95% confidence interval)) | J01A | -- | 1.50 (1.47-1.53) | 2.21 (2.18-2.24) | 2.61 (2.59-2.63) | 2.45 (2.44-2.47) |
| | J01B | -- | 0.01 (0.00-0.01) | 0.00 (0.00-0.01) | 0.00 (0.00-0.00) | 0.00 (0.00-0.00) |
| | J01C | -- | 36.36 (36.23-36.48) | 36.07 (35.98-36.15) | 35.38 (35.31-35.44) | 35.19 (35.15-35.23) |
| | J01D | -- | 27.05 (26.93-27.16) | 28.23 (28.15-28.31) | 28.56 (28.50-28.63) | 30.33 (30.30-30.37) |
| | J01E | -- | 1.49 (1.45-1.52) | 1.30 (1.28-1.32) | 1.29 (1.27-1.30) | 1.46 (1.45-1.47) |
| | J01F | -- | 17.25 (17.15-17.35) | 17.15 (17.08-17.21) | 17.26 (17.20-17.31) | 16.38 (16.35-16.41) |
| | J01G | -- | 1.75 (1.72-1.79) | 1.75 (1.73-1.78) | 1.80 (1.78-1.81) | 2.07 (2.06-2.08) |
| | J01M | -- | 14.22 (14.13-14.31) | 12.85 (12.79-12.91) | 12.55 (12.50-12.59) | 11.41 (11.38-11.44) |
| | J01X | -- | 0.37 (0.36-0.39) | 0.45 (0.44-0.46) | 0.56 (0.55-0.57) | 0.70 (0.69-0.70) |
| | Total antibiotic drug prescriptions (Number) | -- | 564,776 | 1,236,809 | 2,070,703 | 5,204,806 |

A10A: Insulins and Analogues, A10BA: Biguanides, A10BB: Sulfonylureas, A10BD: Combinations of oral blood glucose lowering drugs, A10BF: Alpha glucosidase inhibitors, A10BG: Thiazolidinediones, A10BJ: Glucagon-like peptide-1 (GLP-1) analogues, A10BX: Other blood glucose lowering drugs, excluding insulins, J01A: Tetracyclines, J01B: Amphenicols, J01C: Beta-Lactam Antibacterials, Penicillins, J01D: Other Beta-Lactam Antibacterials, J01E: Sulfonamides and Trimethoprim, J01F: Macrolides, Lincosamides and Streptogramins, J01G: Aminoglycoside Antibacterials, J01M: Quinolone Antibacterials, J01X: Other Antibacterials

The gamma GLM adjusted for age group, antibiotic group, insurance fund, sex, province, and treatment regimen revealed several significant predictors of healthcare costs. Regarding age groups, compared to the reference group of 18−39 years old, older individuals incurred higher costs (Table 5). Specifically, individuals aged 40−64 had 1.33 times higher costs (95% CI: 1.31–1.36), and those aged 65−95 had 1.50 times higher costs (95% CI: 1.47–1.54). Considering antibiotic consumption, increased antibiotic use was associated with progressively higher costs, with the Q4 group incurring 3.15 times the costs of the NAb group (95% CI: 3.07–3.23). Males incurred slightly lower medication costs, with a cost mean ratio of 0.97 (95% CI: 0.96–0.99) compared to females. Among provinces, Zanjan was the only province with significantly higher costs than Tehran (cost mean ratio 1.08, 95% CI: 1.00–1.16, p = 0.044), while Sistan and Baluchestan had the lowest costs at 0.62 times that of Tehran (95% CI: 0.60–0.65) (Fig 3). The treatment regimen was the strongest predictor of costs. Insulins and analogues, when combined with Biguanides, were associated with 5.76 times the mean costs of Biguanides alone (95% CI: 5.55–5.97). Insulins alone resulted in 5.54 times the mean costs

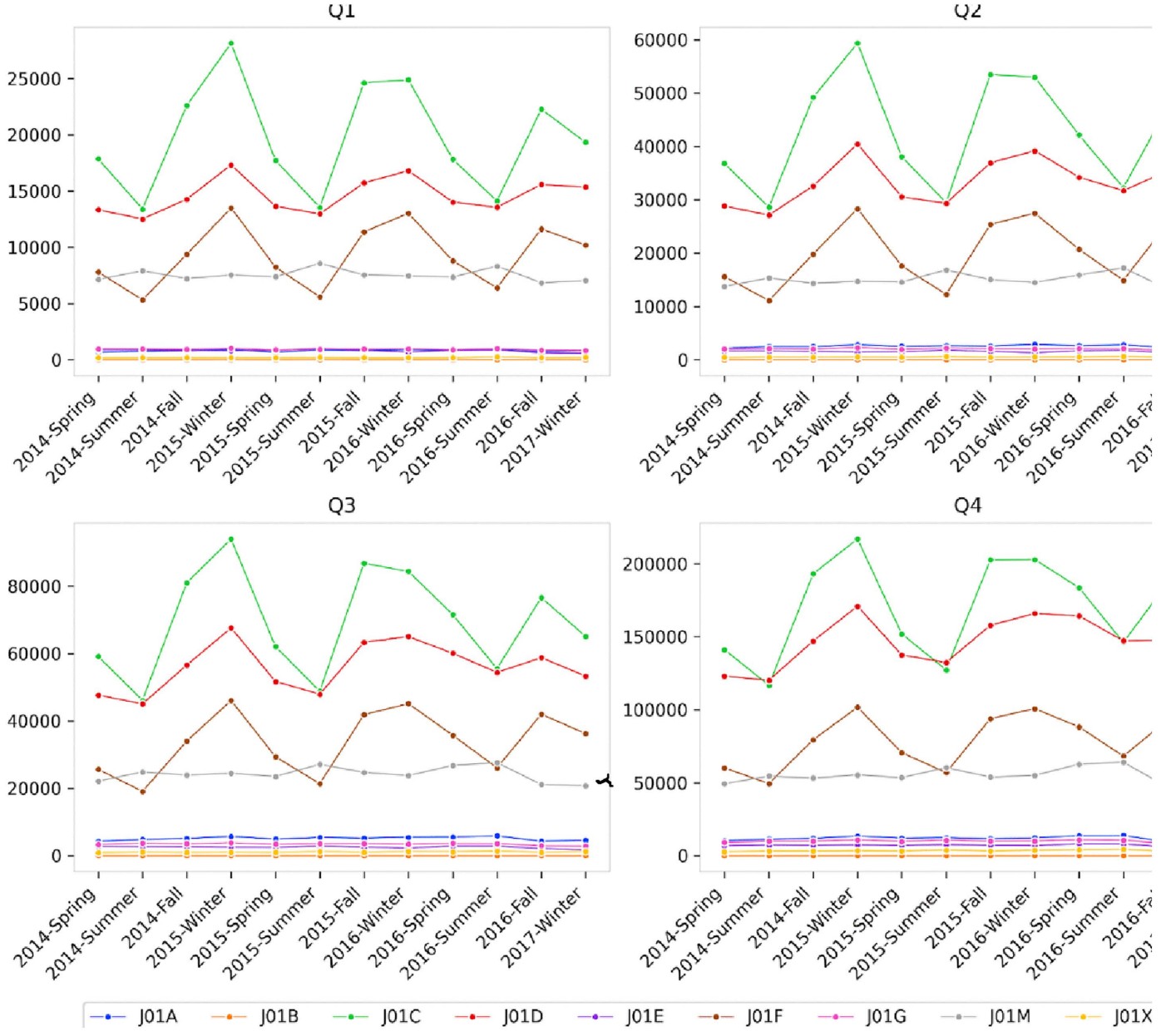

**Fig 2. Seasonal antibiotic drug class prescription frequency by antibiotic group.** J01A: Tetracyclines, J01B: Amphenicols, J01C: Beta-Lactam Antibacterials, Penicillins, J01D: Other Beta-Lactam Antibacterials, J01E: Sulfonamides and Trimethoprim, J01F: Macrolides, Lincosamides and Streptogramins, J01G: Aminoglycoside Antibacterials, J01M: Quinolone Antibacterials, J01X: Other Antibacterials.

of Biguanides (95% CI: 5.39–5.69). The addition of sulfonylureas to insulins and Biguanides was linked to 3.78 times higher mean costs (95% CI: 3.61–3.96), while sulfonylureas alone incurred 0.92 times the mean costs of Biguanides (95% CI: 0.89–0.95). Compared to the fund, all other insurance funds were associated with higher costs, except for Foreign and Universal insurances, which had a lower cost mean ratio (0.63, 95% CI: 0.51–0.77, p = < 0.001 and 0.72, 95% CI: 0.70–0.74, p = < 0.001 respectively). The highest costs were observed in Civil servants and Iranian insurances (S6 and S7 Tables).

**Table 4.  Mean annual costs (% share by category and out-of-pocket).**

| variables | | Antibiotics | | Glucose lowering drugs | | Others | | Total USD (Mean (95% CI)) |
|---|---|---|---|---|---|---|---|---|
| | | Percentage paid out of pocket (95% CI) | Percentage of total cost (95% CI) | Percentage paid out of pocket (95% CI) | Percentage of total cost (95% CI) | Percentage paid out of pocket (95% CI) | Percentage of total cost (95% CI) | |
| Sex | Female | 29.57 (29.48-29.66) | 4.31 (4.30-4.32) | 14.25 (14.17-14.34) | 39.77 (39.48-40.06) | 22.23 (22.00-22.46) | 55.92 (55.26-56.58) | 93 (93-94) |
| | Male | 29.83 (29.65-30.00) | 3.94 (3.92-3.96) | 13.56 (13.45-13.66) | 41.86 (41.46-42.25) | 20.87 (20.56-21.17) | 54.20 (53.34-55.06) | 108 (107-109) |
| Fund | Civil servants | 30.04 (29.93-30.16) | 3.94 (3.93-3.95) | 14.38 (14.28-14.48) | 40.69 (40.34-41.03) | 22.35 (22.05-22.65) | 55.37 (54.57-56.17) | 121 (120-122) |
| | Foreign | 29.91 (27.43-32.39) | 2.11 (1.94-2.28) | 14.02 (13.04-15.01) | 53.71 (48.84-58.58) | 12.73 (11.41-14.04) | 44.18 (30.81-57.55) | 37 (32-43) |
| | Iranian | 29.29 (29.06-29.53) | 3.31 (3.28-3.33) | 13.03 (12.87-13.18) | 37.80 (37.26-38.33) | 17.78 (17.41-18.15) | 58.90 (57.46-60.33) | 115 (113-117) |
| | Others | 30.07 (29.87-30.27) | 6.50 (6.46-6.55) | 14.64 (14.45-14.83) | 33.56 (33.00-34.11) | 24.67 (24.30-25.03) | 59.94 (58.83-61.06) | 119 (118-121) |
| | Rural | 27.97 (27.73-28.20) | 2.81 (2.79-2.83) | 13.29 (13.14-13.45) | 47.96 (47.31-48.61) | 19.92 (19.56-20.29) | 49.24 (48.02-50.45) | 63 (62-64) |
| | Universal | 28.41 (28.22-28.61) | 5.68 (5.64-5.72) | 13.47 (13.30-13.65) | 48.61 (47.82-49.40) | 23.66 (23.21-24.11) | 45.71 (44.55-46.87) | 43 (42-43) |
| Age group | 18-39 | 28.94 (28.80-29.08) | 6.39 (6.36-6.42) | 12.93 (12.76-13.10) | 32.28 (31.78-32.77) | 20.70 (20.38-21.03) | 61.34 (60.06-62.61) | 60 (59-61) |
| | 40-64 | 29.84 (29.73-29.96) | 4.16 (4.15-4.18) | 14.23 (14.14-14.32) | 42.28 (41.96-42.61) | 21.44 (21.18-21.69) | 53.55 (52.82-54.29) | 106 (105-107) |
| | 65-95 | 30.01 (29.80-30.22) | 2.97 (2.94-2.99) | 13.97 (13.86-14.09) | 41.68 (41.25-42.11) | 22.98 (22.61-23.36) | 55.36 (54.44-56.27) | 125 (124-126) |
| Antibiotic group | No antibiotic | 0.00 (0.00) | 0.00 (0.00) | 14.02 (13.86-14.18) | 55.89 (55.11-56.68) | 22.12 (21.54-22.69) | 44.11 (42.77-45.45) | 53 (53-54) |
| | Q1 | 29.98 (29.87-30.09) | 1.33 (1.33-1.34) | 14.18 (14.03-14.32) | 48.01 (47.40-48.61) | 21.58 (21.09-22.07) | 50.66 (49.33-51.99) | 72 (71-73) |
| | Q2 | 29.83 (29.75-29.91) | 2.64 (2.63-2.64) | 14.14 (14.00-14.28) | 43.99 (43.44-44.54) | 21.60 (21.20-22.01) | 53.37 (52.21-54.53) | 90 (88-91) |
| | Q3 | 29.66 (29.59-29.73) | 4.09 (4.08-4.10) | 14.01 (13.86-14.15) | 39.42 (38.92-39.92) | 21.61 (21.22-22.01) | 56.49 (55.33-57.65) | 110 (109-112) |
| | Q4 | 29.60 (29.49-29.71) | 7.67 (7.64-7.69) | 13.71 (13.58-13.85) | 31.26 (30.88-31.63) | 21.88 (21.57-22.19) | 61.08 (60.12-62.03) | 164 (163-166) |

95% CI: 95% confidence interval.

## Discussion

In this nationwide analysis of IHIO claims data, antibiotic use among diabetic patients was common, with Penicillins and other beta-lactams being the most frequently prescribed classes. Costs varied substantially by treatment regimen, sex, and insurance category, with out-of-pocket payments disproportionately higher for antibiotics compared to antidiabetic drugs. Seasonal fluctuations in prescribing highlight potential areas for stewardship interventions. These findings underscore the importance of aligning insurance coverage and prescribing practices with evidence-based guidelines to reduce unnecessary costs and improve patient outcomes.

Sex differences were notable. Women comprised nearly twice the number of men in the study population, consistent with the higher prevalence of diabetes among women in Iran [22,23]. The higher proportion of female patients may reflect biological factors such as abdominal obesity [24–27], as well as social factors including health-seeking behavior. Certain

**Table 5. Median annual costs by category and subgroup.**

| Variables | | Antibiotics (Median (IQR)) | | Glucose lowering drugs (Median (IQR)) | | Others (Median (IQR)) | | Total (Median (IQR)) |
|---|---|---|---|---|---|---|---|---|
| | | Out of pocket | Total* | Out of pocket | Total* | Out of pocket | Total* | |
| **Sex** | **Female** | 0.64 (1.34) | 2.27 (4.50) | 0.80 (2.78) | 3.34 (12.82) | 4.55 (9.13) | 16.29 (32.25) | 29.21 (56.95) |
| | **Male** | 0.56 (1.34) | 2.01 (4.49) | 1.08 (3.75) | 4.82 (18.88) | 3.97 (9.72) | 14.35 (34.52) | 30.88 (68.34) |
| **Fund** | **Civil servants** | 0.85 (1.50) | 2.95 (4.95) | 1.22 (4.49) | 5.13 (20.10) | 6.31 (10.93) | 22.23 (38.10) | 39.80 (69.60) |
| | **Foreign** | 0.00 (0.30) | 0.13 (1.03) | 1.18 (2.05) | 5.31 (13.10) | 0.95 (1.73) | 3.27 (6.10) | 11.79 (21.55) |
| | **Iranian** | 0.65 (1.22) | 2.29 (4.09) | 0.93 (3.25) | 3.96 (16.08) | 4.38 (8.42) | 15.77 (30.67) | 30.20 (60.10) |
| | **Others** | 1.29 (2.43) | 4.49 (8.02) | 1.05 (3.69) | 4.40 (16.90) | 8.92 (15.78) | 31.09 (54.17) | 51.94 (85.73) |
| | **Rural** | 0.18 (0.62) | 0.69 (2.20) | 0.66 (1.74) | 2.65 (8.74) | 2.04 (4.73) | 7.42 (17.34) | 15.80 (34.14) |
| | **Universal** | 0.38 (0.91) | 1.40 (2.99) | 0.68 (1.46) | 2.65 (6.98) | 1.84 (3.53) | 6.72 (12.92) | 14.89 (25.60) |
| **Age group** | **18-39** | 0.60 (1.25) | 2.16 (4.35) | 0.49 (0.93) | 2.01 (4.21) | 2.42 (5.19) | 8.70 (18.82) | 17.58 (31.74) |
| | **40-64** | 0.66 (1.44) | 2.34 (4.86) | 1.13 (3.94) | 4.84 (18.21) | 4.62 (9.08) | 16.40 (31.75) | 32.16 (62.43) |
| | **65-95** | 0.54 (1.18) | 1.89 (3.96) | 1.53 (5.19) | 6.67 (24.83) | 7.72 (13.65) | 27.74 (48.17) | 47.40 (85.67) |
| **Antibiotic group** | **No antibiotic** | 0.00 (0.00) | 0.00 (0.00) | 0.80 (2.18) | 3.11 (10.55) | 1.37 (3.75) | 4.78 (13.09) | 11.28 (29.51) |
| | **Q1** | 0.25 (0.20) | 0.83 (0.66) | 0.80 (2.72) | 3.45 (12.80) | 2.64 (5.93) | 9.35 (20.98) | 18.15 (40.83) |
| | **Q2** | 0.65 (0.37) | 2.20 (1.15) | 0.89 (3.15) | 3.78 (14.79) | 3.97 (7.56) | 14.16 (26.79) | 25.66 (49.49) |
| | **Q3** | 1.26 (0.64) | 4.27 (1.90) | 0.96 (3.51) | 4.11 (16.59) | 5.78 (9.47) | 20.62 (33.56) | 35.73 (59.54) |
| | **Q4** | 2.80 (2.20) | 9.55 (6.86) | 1.08 (4.10) | 4.68 (19.83) | 10.86 (15.48) | 38.49 (54.01) | 65.40 (91.95) |

*The sum of out-of-pocket and insurance-paid cost,

IQR = Interquartile range.

The numbers are in USD

antidiabetic agents are also prescribed in conditions such as obesity and polycystic ovarian syndrome, which may contribute to higher drug use among women [23,28–30]. Costs, however, were higher in men, suggesting that sex differences in both disease burden and healthcare utilization warrant further investigation. Age also influenced outcomes: the largest group was middle-aged adults (40–64 years), while the elderly group (65–95 years) was smaller, likely reflecting higher mortality [22,23]. Increasing age was associated with higher costs, consistent with greater comorbidity and complications.

Across all groups, beta-lactams were the most prescribed antibiotics, particularly Penicillins. Their frequent use is consistent with empirical treatment practices [26,31] and the prevalence of gram-positive pathogens in diabetic infections such as diabetic foot [27,29,32,33]. However, as antibiotic use increased, the proportion of Penicillins declined while other beta-lactams and aminoglycosides increased, possibly reflecting treatment escalation in patients with more complicated infections. This pattern highlights the need for careful monitoring of antibiotic use in diabetic populations, who are at higher risk of multidrug-resistant organisms [30,34]. Prior studies indicate that cumulative antibiotic exposure is a predictor of MDR development [31,35]. Over-the-counter antibiotic use, which is prevalent in Iran and other developing countries [32,33,36,37], may also contribute to inappropriate prescribing. Poor glucose control has been identified as another risk factor for MDR infections [33,34]

Higher antibiotic use was a major predictor of total drug costs, even though antibiotics themselves accounted for a relatively small share of overall expenditures. Out-of-pocket payments for antibiotics were disproportionately high, suggesting lower insurance coverage compared to antidiabetic drugs [30,34,35]. This finding emphasizes the importance of accurate first-line antibiotic therapy and adequate insurance coverage to reduce repeated consultations and treatment failures. The proportion of insulin prescriptions increased in groups with higher antibiotic use, consistent with more advanced or uncontrolled diabetes. Since insulin use was also associated with higher costs, this may explain why patients with greater antibiotic exposure incurred higher overall expenditures. The low use of newer agents such as SGLT-2 inhibitors reflects their limited availability in Iran during the study period.

| | | |
|---|---|---|
| **Age group** | | |
| 65-95 | 1.50(1.47 to 1.54) | |
| 40-64 | 1.33(1.31 to 1.36) | |
| **Antibiotic group** | | |
| Q4 | 3.17(3.09 to 3.25) | |
| Q3 | 2.12(2.07 to 2.17) | |
| Q2 | 1.71(1.67 to 1.75) | |
| Q1 | 1.39(1.36 to 1.42) | |
| **Dominant diabetes treatment regimen** | | |
| A10A A10BA | 5.75(5.54 to 5.97) | |
| A10A | 5.53(5.38 to 5.68) | |
| A10A A10BA A10BB | 3.78(3.61 to 3.96) | |
| other | 2.32(2.25 to 2.39) | |
| A10BA A10BB A10BF | 1.81(1.71 to 1.92) | |
| A10BA A10BB A10BG | 1.65(1.55 to 1.74) | |
| A10BA A10BG | 1.43(1.33 to 1.54) | |
| A10BG | 1.26(1.17 to 1.36) | |
| A10BA A10BB | 1.08(1.06 to 1.11) | |
| A10BB | 0.92(0.89 to 0.95) | |
| **Fund** | | |
| Civil servants | 1.51(1.48 to 1.55) | |
| Iranian | 1.51(1.47 to 1.55) | |
| Others | 1.41(1.37 to 1.46) | |
| Universal | 0.72(0.70 to 0.74) | |
| Foreign | 0.63(0.51 to 0.77) | |
| **Gender** | | |
| Male | 0.98(0.96 to 0.99) | |
| **Province** | | |
| Zanjan | 1.08(1.00 to 1.16) | |
| Gilan | 0.99(0.96 to 1.03) | |
| Fars | 0.92(0.90 to 0.95) | |
| Yazd | 0.90(0.85 to 0.95) | |
| Kerman | 0.89(0.86 to 0.93) | |
| Isfahan | 0.89(0.87 to 0.92) | |
| Qazvin | 0.85(0.79 to 0.91) | |
| Chaharmahal and Bakhtiari | 0.83(0.78 to 0.89) | |
| Khorasan, Razavi | 0.83(0.81 to 0.85) | |
| Hamadan | 0.82(0.78 to 0.86) | |
| Markazi | 0.81(0.76 to 0.85) | |
| Mazandaran | 0.78(0.75 to 0.81) | |
| Bushehr | 0.77(0.72 to 0.82) | |
| Golestan | 0.75(0.72 to 0.78) | |
| Kermanshah | 0.75(0.72 to 0.78) | |
| Kurdistan | 0.71(0.68 to 0.74) | |
| Khorasan, South | 0.70(0.65 to 0.75) | |
| Hormozgan | 0.68(0.64 to 0.72) | |
| Khorasan, North | 0.67(0.63 to 0.72) | |
| Lorestan | 0.66(0.63 to 0.69) | |

**Fig 3. Adjusted mean ratios (95% CI) from Gamma GLM for total costs.** Reference values of variables are: 18−39 for age group, No antibiotic for antibiotic group, A10BA for dominant diabetes treatment regimen, rural for fund, female for sex, and Tehran for province. A10A: Insulins and Analogues, A10BA: Biguanides, A10BB: Sulfonylureas, A10BD: Combinations of oral blood glucose lowering drugs, A10BF: Alpha glucosidase inhibitors, A10BG: Thiazolidinediones, A10BJ: Glucagon-like peptide-1 (GLP-1) analogues, A10BX: Other blood glucose lowering drugs, excluding insulins.

This study was limited by its retrospective nature and reliance on prescription claims data, which do not include clinical diagnoses or patient outcomes. Although the approach used in this study has been previously validated for population-level estimates using administrative data, its high specificity and limited sensitivity require that the results be interpreted with caution [17,20]. Additionally, the lack of OTC prescription data and data from some provinces may limit the generalizability of the findings to the entire Iranian population. Furthermore, although our study covered approximately 32% of the diabetic population in Iran, the observed patterns may differ among individuals insured by other insurance providers. Our study was also limited by the lack of data on prescriptions during potential hospitalizations. However, the current study was of high value regarding its topic and the large sample size. Also, the data used in the study was safe from recall bias as it was used the recorded data. Future research should aim to integrate clinical data and examine the long-term outcomes of patients receiving antibiotics, as well as the effectiveness of various antibiotic stewardship interventions in diabetic populations.

Future research should evaluate the appropriateness of antibiotic use in diabetic patients, explore cost-effectiveness of different regimens, and assess interventions to optimize prescribing practices. Integrating clinical data with claims information would allow examination of long-term outcomes and strengthen stewardship efforts in this vulnerable population.

## Conclusions

It is concluded that the total cost of treatment for diabetic patients is strongly associated with the anti-diabetic and anti-biotic regimens they receive. Additionally, there are variations in prescription patterns between men and women, and treatment costs change as patients age. These findings are valuable as they can inform policies regarding antibiotic therapy in DM patients, insurance coverage, and patient education plans. However, further studies are needed to assess patient outcomes after each regimen. Infection among DM patients is a complex condition that requires accurate diagnosis and prescription by physicians, and policy makers create conditions for good patient compliance.

## Supporting information

**S1 Table. The percentage of included participants from different provinces in the study.**
(DOCX)

**S2 Table. Percentage of dominant diabetes treatment regimens for antibiotic groups.** A10A: Insulins and Analogues, A10BA: Biguanides, A10BB: Sulfonylureas, A10BD: Combinations of oral blood glucose lowering drugs, A10BF: Alpha glucosidase inhibitors, A10BG: Thiazolidinediones, A10BJ: Glucagon-like peptide-1 (GLP-1) analogues, A10BX: Other blood glucose lowering drugs, excl. insulins.
(DOCX)

**S3 Table. Seasonal trends in antibiotic class prescriptions (2014–2017).** J01A: Tetracyclines, J01B: Amphenicols, J01C: Beta-Lactam Antibacterials, Penicillins, J01D: Other Beta-Lactam Antibacterials, J01E: Sulfonamides and Tri-methoprim, J01F: Macrolides, Lincosamides and Streptogramins, J01G: Aminoglycoside Antibacterials, J01M: Quinolone Antibacterials, J01X: Other Antibacterials.
(DOCX)

**S4 Table. Median annual costs by province.**
(DOCX)

**S5 Table. Mean annual costs by province.**
(DOCX)

**S6 Table. Gamma Generalized linear model (GLM) adjusted mean ratios for total medication costs.** Reference values of variables are: 18−39 for age group, No antibiotic for antibiotic group, A10BA for dominant diabetes treatment

regimen, rural for fund, female for sex, and Tehran for province. A10A: Insulins and Analogues, A10BA: Biguanides, A10BB: Sulfonylureas, A10BD: Combinations of oral blood glucose lowering drugs, A10BF: Alpha glucosidase inhibitors, A10BG: Thiazolidinediones, A10BJ: Glucagon-like peptide-1 (GLP-1) analogues, A10BX: Other blood glucose lowering drugs, excl. insulins.
(DOCX)

**S7 Table. Gamma Generalized linear model (GLM) crude mean ratios for total medication costs.** Reference values of variables are: 18−39 for age group, No antibiotic for antibiotic group, A10BA for dominant diabetes treatment regimen, rural for fund, female for sex, and Tehran for province. A10A: Insulins and Analogues, A10BA: Biguanides, A10BB: Sulfonylureas, A10BD: Combinations of oral blood glucose lowering drugs, A10BF: Alpha glucosidase inhibitors, A10BG: Thiazolidinediones, A10BJ: Glucagon-like peptide-1 (GLP-1) analogues, A10BX: Other blood glucose lowering drugs, excl. insulins.
(DOCX)

## Author contributions

**Data curation:** Arash Bagherian Ghotbi, Benyamin Khoshparast, Hamidreza Hekmat, Zahra Shahali.

**Methodology:** Arash Bagherian Ghotbi, Ali Golestani.

**Supervision:** Ozra Tabatabaei-Malazy.

**Validation:** Arash Bagherian Ghotbi, Ali Golestani, Ozra Tabatabaei-Malazy.

**Writing – original draft:** Benyamin Khoshparast.

**Writing – review & editing:** Arash Bagherian Ghotbi, Hamidreza Hekmat, Zahra Shahali, Ali Golestani, Ozra Tabatabaei-Malazy.

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
