## [Decision Letter · Decision Letter 0]

17 Nov 2025

Dear Dr.  Tabatabaei-Malazy,

Thank you for submitting your manuscript to PLOS ONE. After careful consideration, we feel that it has merit but does not fully meet PLOS ONE’s publication criteria as it currently stands. Therefore, we invite you to submit a revised version of the manuscript that addresses the points raised during the review process

We look forward to receiving your revised manuscript.

Kind regards,

Marwan Salih Al-Nimer, MD, PhD

Academic Editor

PLOS ONE

Journal Requirements:

Reviewer's Responses to Questions

**Comments to the Author**

1. Is the manuscript technically sound, and do the data support the conclusions?

Reviewer #1: No

Reviewer #2: Yes

2. Has the statistical analysis been performed appropriately and rigorously?

Reviewer #1: I Don't Know

Reviewer #2: Yes

3. Have the authors made all data underlying the findings in their manuscript fully available?

Reviewer #1: No

Reviewer #2: No

4. Is the manuscript presented in an intelligible fashion and written in standard English?

Reviewer #1: No

Reviewer #2: Yes

Reviewer #1: This manuscript addresses the important issue of increasing prevalence of diabetes and, from the abstract, seeks to evaluate the primary outcomes of antibiotic consumption and medication costs in diabetic patients. The authors use an available data set to retrospectively analyze these variables. The methodology selected includes simple descriptive statistics of means of cumulative days of antibiotic use and drug costs, and Generalized Linear Models to evaluate relationships among variables.

The methodology seems to have included any patient receiving a diabetes medication at any point in time within the dataset, leading to patients who could have received such medication beginning years before the study to patients who had received such medication for a only single month at the end of the study period. The authors seem to have considered these differences only when calculating the mean annual cost of diabetes medications (Line 202-203).

My understanding is that a different methodology and statistical analysis should have been used to account for the above. With the current methodology the findings are unclear and of questionable value.

I have indicated that statistical and methodological review would be useful.

I have also indicated that the authors have stated that all data cannot be made available due to specific limitations.

As I have also indicated that the manuscript is not presented in an intelligible fashion, I offer the following comments for consideration once review of the methodology / statistical analysis and revisions have been completed:

1. The introduction includes a range of topics that seem unrelated to the purpose of the manuscript. For example, non-adherence to medications, complexity of medication regimes and poly pharmacy. The introduction jumps from topic to topic, providing single sentences on each. This does not create a compelling rationale for the completion of the study.

2. Authors used a dataset available from a single insurance organization from a section of Iranian provinces covering the period of 2014-2017. No information is provided as to how representative the sample in the data set is of the Iranian population. Potential biases that are not addressed include differences among states, insurance companies and categories within the insurance company. This lack of information is important because the authors analyze results according to states, categories of insurance, out of pocket costs etc. Without explanation, the rationale for these analyses and results cannot be understood.

3. The methodology for identifying diabetics is described under ‘data processing and analysis’ (lines 167-171), listing reference 21 as the source for their methodology. However, reference 21 identifies specifically that using only dispensings of diabetic drugs is a specific but not overly sensitive method. This limitation is not addressed by the authors.

4. The authors do not identify primary or second outcomes explicitly. Nor are modifying variables thoroughly listed and rationalized. The authors introduce a range of variables including age, sex, province, dominant diabetes treatment regimen (lines 185-195), and insulin/ no insulin status. Additional variables are reported (e.g. Civil servants vs rural insurance, seasonal variation, out-of-pocket vs insurance) without explaining the rationale or how categorized. This makes it difficult to understand the results.

5. Results: as presented are confusing and difficult to follow. Readers are referred to multiple tables in the manuscript and supplemental information. The results should be organized according to standard format of demographics followed by results on primary outcomes. Context should be provided and not just a series of sentences listing results.

6. Tables:

i. Require labels (e.g. Table one are percent and 95% confidence Intervals, Table S1 the first two rows are % while the last row is total count), add ‘,’ to facilitate reading of large numbers.

ii. Should be shortened and information from S1 brought forward to main manuscript.

iii. The difference between, and value of, Tables 2 and 3 are unclear. Manuscript on line 279 refers to Table 3 as cost data and it is not.

iv. Tables and figures that include variables not described in the background or methodology should be excluded (state, insurance category, seasonal, out of pocket).

7. Discussion: extrapolates beyond the purpose and findings of this study, introducing a range of topics not previously addressed.

Reviewer #2: - The study methods are appropriate and reproducible.

- The study met the applicable standards regarding research ethics.

- Statistical analyses were conducted in accordance with the research requirements and the type of data.

- Line 234, instead of "Baseline characteristics …" the title of table (1) should be rewriting to clarify that values are percentages of the prescriptions among groups.

- Also, in the same table (1), I suggest adding row for each variable to list the sum of the percentages.

- On line 236, the sentence should be rephrased so that it does not begin with the number.

- During the sentence “The proportion ….”, which begins on line 236, the table number that displays the data must be indicated.

- I suggest that "… and 95% Confidence Intervals …" be deleted from all tables headings (e.g; lines 263 & 294), and referred to in the text.

- Legends of Tables (4&5), and Fig. (3) are too long, so shorten it if possible.

**Do you want your identity to be public for this peer review?** For information about this choice, including consent withdrawal, please see our Privacy Policy

Reviewer #1: No

Reviewer #2: **Yes:** ABDULRAZZAQ YAHYA AHMED AL-KHAZZAN

---

## [Author Response · Author response to Decision Letter 1]

20 Dec 2025

Response to Reviewers

Authors: We thank the Academic Editor and Reviewers for their careful evaluation of our manuscript and for the constructive comments provided. We have revised the manuscript thoroughly to address each point raised. Below we provide a detailed, point‑by‑point response.

Editorial Requirements

1.Please ensure that your manuscript meets PLOS ONE's style requirements, including those for file naming. The PLOS ONE style templates can be found at:

Response: Thank you for your feedback. Formatting and Style We have reformatted the manuscript according to the PLOS ONE style templates for both the main body and the title/authors/affiliations. File naming has been updated to comply with journal requirements.

Response: Thank you for your feedback. All funding details have been removed from the manuscript text. Funding information will be provided only in the Funding Statement section of the online submission form.

3. We note that you have indicated that there are restrictions to data sharing for this study. PLOS only allows data to be available upon request if there are legal or ethical restrictions on sharing data publicly. For more information on unacceptable data access restrictions

Response: Thank you for your feedback. We have revised the Data Availability Statement to comply with PLOS ONE policy. The Iran Health Insurance Organization (IHIO) owns the claims dataset analyzed, and it contains potentially identifying patient information. However, any researcher with written permission can request to obtain the anonymized data. Requests for access to a de‑identified dataset may be sent to IHIO Data Access Committee (https://nchir.ihio.gov.ir/).

Response: Thank you for your feedback. Ethics Statement The ethics statement has been moved to the Methods section only and removed from other sections.

5. Please include captions for your Supporting Information files at the end of your manuscript, and update any in-text citations to match accordingly.

Response: Supporting Information Captions for all Supporting Information files have been added at the end of the manuscript, and in‑text citations have been updated accordingly.

Response: Reviewer‑Suggested Citations We have reviewed the suggested works and included citations where relevant.

Reviewer 1 Comments

This manuscript addresses the important issue of increasing prevalence of diabetes and, from the abstract, seeks to evaluate the primary outcomes of antibiotic consumption and medication costs in diabetic patients. The authors use an available data set to retrospectively analyze these variables. The methodology selected includes simple descriptive statistics of means of cumulative days of antibiotic use and drug costs, and Generalized Linear Models to evaluate relationships among variables.

The methodology seems to have included any patient receiving a diabetes medication at any point in time within the dataset, leading to patients who could have received such medication beginning years before the study to patients who had received such medication for a only single month at the end of the study period. The authors seem to have considered these differences only when calculating the mean annual cost of diabetes medications (Line 202-203).

My understanding is that a different methodology and statistical analysis should have been used to account for the above. With the current methodology the findings are unclear and of questionable value.

I have indicated that statistical and methodological review would be useful.

I have also indicated that the authors have stated that all data cannot be made available due to specific limitations.

Response: The authors would like to express their most sincere words of appreciation for the time and kind consideration of the reviewer. Thank you for your thoughtful comments, which we believe have significantly improved the quality of our work.

Regarding the methodology, as we explained, we included all patients with at least one prescription for antidiabetic medication in our analysis. This approach has been used in previous studies to estimate diabetes prevalence at the population level (for example, in the study by Huber et al., “Identifying patients with chronic conditions using pharmacy data in Switzerland: an updated mapping approach to the classification of medications.”). We then applied methods to identify each patient’s dominant antidiabetic regimen and to classify antibiotic use groups. It is important to note that we did not have information on patients prior to the study period. Data were available only from 2014 to 2017. Therefore, the maximum duration a patient could be identified as having diabetes in our database was three years.

To calculate medication costs, we reported annual costs to adjust for the cumulative effect of time. For each patient, we calculated the total medication costs for the years they were identified as diabetic in the dataset. For example, for a patient detected in the first year, we calculated costs for the first, second, and third years, and the average was considered the annual mean cost. For patients detected in the second year, only costs from the second and third years were included, and for those detected in the third year, only the third year costs were considered. As noted, patients identified toward the end of any given year (not only the last year) may show lower costs for that year. Newly diagnosed patients generally incur lower costs compared with patients with long-term diabetes. When reporting population-level estimates and summarizing all diabetic patients in the population, it is appropriate to include all patients regardless of the duration of their diabetes.

1. The introduction includes a range of topics that seem unrelated to the purpose of the manuscript. For example, non-adherence to medications, complexity of medication regimes and poly pharmacy. The introduction jumps from topic to topic, providing single sentences on each. This does not create a compelling rationale for the completion of the study.

Response: Thank you for your meticulous comment. We have streamlined the Introduction to focus on diabetes prevalence, infection risk, and antibiotic/cost burden. Tangential topics such as non‑adherence and polypharmacy have been removed unless directly relevant.

2. Authors used a dataset available from a single insurance organization from a section of Iranian provinces covering the period of 2014-2017. No information is provided as to how representative the sample in the data set is of the Iranian population. Potential biases that are not addressed include differences among states, insurance companies and categories within the insurance company. This lack of information is important because the authors analyze results according to states, categories of insurance, out of pocket costs etc. Without explanation, the rationale for these analyses and results cannot be understood.

Response: Thank you for your thoughtful comment. We added a detailed explanation of the IHIO and its insurance coverage in Iran. Based on our diabetic study sample and estimates of the diabetic population in Iran in 2016 (the year overlapping with our study period) we determined that approximately 32% of all individuals with diabetes in Iran were included in our analysis. In addition, we provided further clarification of the study variables, particularly the different insurance funds, to improve readers’ understanding.

3. The methodology for identifying diabetics is described under ‘data processing and analysis’ (lines 167-171), listing reference 21 as the source for their methodology. However, reference 21 identifies specifically that using only dispensings of diabetic drugs is a specific but not overly sensitive method. This limitation is not addressed by the authors.

Response: Thank you for your accurate comment. The approach used in this study has been applied in several previous studies such as “Identifying patients with chronic conditions using pharmacy data in Switzerland: an updated mapping approach to the classification of medications” by Huber et al. It is a validated method for estimating disease prevalence at the population level using administrative data in the absence of diagnostic codes. However, as you noted, this approach may overestimate the prevalence of diabetes because of its high specificity and limited sensitivity. We have acknowledged this limitation by explicitly addressing it both in the Methods section, where the approach is described, and in the Limitations subsection of the Discussion.

4. The authors do not identify primary or second outcomes explicitly. Nor are modifying variables thoroughly listed and rationalized. The authors introduce a range of variables including age, sex, province, dominant diabetes treatment regimen (lines 185-195), and insulin/ no insulin status. Additional variables are reported (e.g. Civil servants vs rural insurance, seasonal variation, out-of-pocket vs insurance) without explaining the rationale or how categorized. This makes it difficult to understand the results.

Response: Thank you for your accurate comment. We added the following explanation at the end of the first paragraph of the Methods section. In this study, we first aimed to describe the prescribing patterns of antidiabetic and antibiotic medications across different levels of antibiotic use. We then assessed medication costs across the available sociodemographic and clinical subgroups represented in the data:

“This study had two main outcomes. First, it described the prescription patterns of antidiabetic and antibiotic medications across different groups of antibiotic use. Second, it compared the costs of antidiabetic medications, antibiotics, and other drugs across available sociodemographic and clinical variables”.

5. Results: as presented are confusing and difficult to follow. Readers are referred to multiple tables in the manuscript and supplemental information. The results should be organized according to standard format of demographics followed by results on primary outcomes. Context should be provided and not just a series of sentences listing results.

Response: Thank you for your meticulous comment. We made several changes in the Results section to improve clarity and prevent confusion. First, we added headings for each subsection. In the first subsection, we described the study population and the prescriptions included. Next, we presented the prescribing patterns of antidiabetic medications across different antibiotic use groups. This was followed by a section about antibiotic prescribing patterns across groups and over time. Finally, we reported both descriptive and model-based results of medication costs according to the available sociodemographic and clinical variables. We also removed Table 3, as its results did not provide additional value or meaningful insights and could potentially confuse readers.

6. Tables:

i. Require labels (e.g. Table one are percent and 95% confidence Intervals, Table S1 the first two rows are % while the last row is total count), add ‘,’ to facilitate reading of large numbers.

ii. Should be shortened and information from S1 brought forward to main manuscript.

iii. The difference between, and value of, Tables 2 and 3 are unclear. Manuscript on line 279 refers to Table 3 as cost data and it is not.

iv. Tables and figures that include variables not described in the background or methodology should be excluded (state, insurance category, seasonal, out of pocket).

Response: Thank you for your meticulous comment. Labels and titles of tables and figures were reviewed and updated, and any ambiguities were clarified. Commas were added to large numbers in the tables for improved readability. Table S1 was moved to the main manuscript as Table 2. All citations to tables and figures were checked, and any errors were corrected. Additionally, all variables reported in the Results section are now fully described and referenced in the Methods section.

7. Discussion: extrapolates beyond the purpose and findings of this study, introducing a range of topics not previously addressed.

Response: We appreciate this insightful comment. The Discussion section has been thoroughly revised to remain focused on the study’s actual findings. Extraneous material and extrapolations beyond the scope of the data have been removed, and the section now emphasizes results directly supported by our analyses, along with clearly stated limitations and relevant future research directions.

Reviewer 2 Comments

The study methods are appropriate and reproducible.

- The study met the applicable standards regarding research ethics.

- Statistical analyses were conducted in accordance with the research requirements and the type of data.

Authors: The authors would like to express their most sincere words of appreciation for the time and kind consideration of the reviewer. Thank you for your thoughtful comments, which we believe have significantly improved the quality of our work.

1. Line 234, instead of "Baseline characteristics …" the title of table (1) should be rewriting to clarify that values are percentages of the prescriptions among groups.

Response: Thank you for your accurate comment. Table 1 shows the percentage of participants stratified by different subgroups (age groups, sex, so on), in each quartile of antibiotic consumption. For example, in column ‘All’, 26.02% for 18-39 age group shows that 26.02% of included participants were 18-39 years old. We changed the title to “The percentage of included participants based on demographic and insurance stratifications in the study” to make this clearer.

2. Also, in the same table (1), I suggest adding row for each variable to list the sum of the percentages.

Response: Thank you for your accurate comment. Sums of different subgroups of each stratification is equal to 100% (equal to all participants included in the study). We added this sentence in result section to make this clearer: “Percentages within each demographic and insurance stratifications sum to 100%, and are presented with 95% confidence intervals.”

3. On line 236, the sentence should be rephrased so that it does not begin with the number.

Response: Thank you for your accurate comment. Line 236 sentence rephrased to: “Of all prescriptions, 7.79% contained anti‑diabetic drugs ….”

4. During the sentence “The proportion ….”, which begins on line 236, the table number that displays the data must be indicated.

Response: Thank you for your accurate comment. We checked all the citations to Tables and Figures and completely updated them.

5. I suggest that "… and 95% Confidence Intervals …" be deleted from all tables headings (e.g; lines 263 & 294), and referred to in the text.

Response: Thank you for your accurate comment. Removed “and 95% Confidence Intervals” from table headings; confidence intervals are mentioned in the text.

6. Legends of Tables (4&5), and Fig. (3) are too long, so shorten it if possible.

Response: Thank you for your accurate comment. Legends for Tables 4 & 5 and Figure 3 have been shortened.

Authors: The manuscript has been thoroughly proofread for grammar and readability. Long sentences have been simplified, terminology standardized (e.g., “Insulins and analogues”), and clarity imp

---

## [Editor Report · Decision Letter 1]

9 Jan 2026

Dear Dr. Tabatabaei-Malazy,

Thank you for submitting your manuscript to PLOS ONE. After careful consideration, we feel that it has merit but does not fully meet PLOS ONE’s publication criteria as it currently stands. Therefore, we invite you to submit a revised version of the manuscript that addresses the points raised during the review process.

**ACADEMIC EDITOR: Minor revision**

We look forward to receiving your revised manuscript.

Kind regards,

Marwan Salih Al-Nimer, MD, PhD

Academic Editor

PLOS One

Journal Requirements:

Additional Editor Comments:

References should be typed according to the PLoS ONE guidelines.

---

## [Author Response · Author response to Decision Letter 2]

25 Jan 2026

Response to Reviewers

Authors: We thank the Academic Editor and Reviewers for their careful evaluation of our manuscript and for the constructive comments provided. We have revised the manuscript thoroughly to address each point raised.

Editorial Requirements

References should be typed according to the PLoS ONE guidelines.

Response: Thank you for your feedback. Your comment is considered.

Sincerely,

Authors

---

## [Editor Report · Decision Letter 2]

1 Feb 2026

Antibiotic Consumption and Medication Cost in Diabetic Patients: Insights from Iran Health Insurance Organization (IHIO) Claims Data

PONE-D-25-54928R2

Dear Dr. Ozra Tabatabaei-Malazy,

We’re pleased to inform you that your manuscript has been judged scientifically suitable for publication and will be formally accepted for publication once it meets all outstanding technical requirements.

Kind regards,

Marwan Salih Al-Nimer, MD, PhD

Academic Editor

PLOS One
---

## [Editor Report · Acceptance letter]

PONE-D-25-54928R2

PLOS One

Dear Dr. Tabatabaei-Malazy,

I'm pleased to inform you that your manuscript has been deemed suitable for publication in PLOS One. Congratulations! Your manuscript is now being handed over to our production team.

Kind regards,

on behalf of

Professor Marwan Salih Al-Nimer

Academic Editor

PLOS One